# Accelerating Training of Batch Normalization: A Manifold Perspective

**Mingyang Yi** [*1,2]

[1]University of Chinese Academy of Sciences
[2]Academy of Mathematics and Systems Science, Chinese Academy of Sciences

## Abstract

Batch normalization (BN) has become a critical component across diverse deep neural networks. The network with BN is invariant to positively linear re-scale transformation, which makes there exist infinite functionally equivalent networks with different scales of weights. However, optimizing these equivalent networks with the first-order method such as stochastic gradient descent will obtain a series of iterates converging to different local optima owing to their different gradients across training. To obviate this, we propose a quotient manifold *PSI manifold*, in which all the equivalent weights of the network with BN are regarded as the same element. Next, we construct gradient descent and stochastic gradient descent on the proposed PSI manifold to train the network with BN. The two algorithms guarantee that every group of equivalent weights (caused by positively re-scaling) converge to the equivalent optima. Besides that, we give convergence rates of the proposed algorithms on the PSI manifold. The results show that our methods accelerate training compared with the algorithms on the Euclidean weight space. Finally, empirical results verify that our algorithms consistently improve the existing methods in both convergence rate and generalization ability under various experimental settings.

## 1 INTRODUCTION

Batch normalization (BN) [Ioffe and Szegedy, 2015] is one of the most critical innovations in deep learning, which appears to help optimization as well as generalization [Ioffe and Szegedy, 2015, Santurkar et al., 2018]. Despite the success of BN, there is one adverse effect in terms of op-

timization due to the positively scale-invariant (PSI) property brought by it. The PSI property is explained as the weights in each layer with BN are invariant to positively linear re-scaling. Due to this, there can be an infinite number of networks functionally equivalent to each other but with various scales of weights. These networks can converge to different local optima owing to different gradients [Cho and Lee, 2017, Huang et al.]. Then the converged point can be sensitive to the scale of parameters, and some of them may have poor performances (see Section 4). Hence, it is desirable to obviate such ambiguity of training.

To this end, we leverage the technique of optimization on manifold [Absil et al., 2009, Cho and Lee, 2017, Huang et al., Badrinarayanan et al., 2015]. We propose to constrain the scale-invariant weights of deep neural networks with BN in a quotient manifold i.e. *PSI manifold* defined in Section 4, in which all the positively scale-equivalent weights are viewed as the same element. Constraining the scale-invariant weights on the PSI manifold maintains the representation ability of hypothesis space due to the PSI property of network with BN. More importantly, optimizing on the manifold breaks the training ambiguity caused by the PSI property.

By constructing the Riemannian metric and retraction function [Absil et al., 2009] on such PSI manifold [Boumal et al., 2019], we propose the gradient descent (GD), stochastic gradient descent (SGD), and SGD with momentum on such manifold. We abbreviate the corresponding algorithms as PSI-GD, PSI-SGD, and PSI-SGDM, respectively. Compared with vanilla GD and SGD on the Euclidean weight space, the proposed algorithms guarantee the equivalent scale-invariant weights (caused by positively re-scaling) converge to the equivalent local optima.

In contrast to the literature of optimizing network with BN while constraining parameters on manifold [Cho and Lee, 2017, Huang et al., Badrinarayanan et al., 2015,?], we give the convergence rates of our PSI-GD and PSI-SGD. The convergence rates of the two methods are respectively in

---
[*]yimingyang17@mails.ucas.edu.cn

*Accepted for the 38th Conference on Uncertainty in Artificial Intelligence* (UAI 2022).

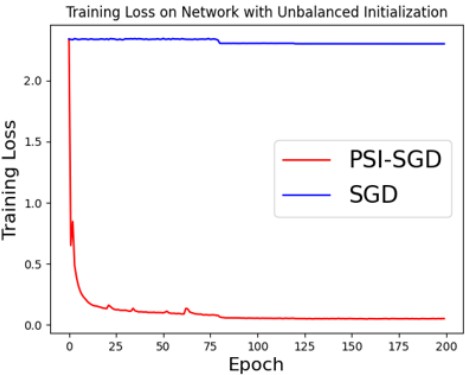
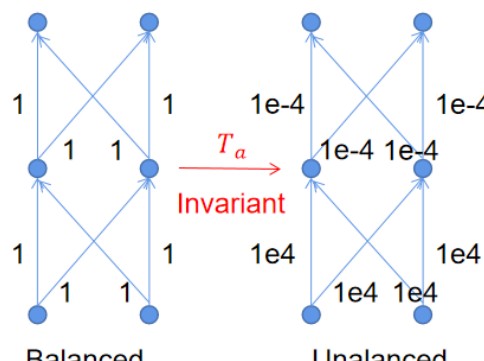

Figure 1: The figure on the right hand side are two batch normalized two-layer neural networks with weights $W$ and $T_a(W)$, where $W$ and $T_a(W)$ are respectively balanced and unbalanced weights. The PSI property says that $f(x, W) = f(x, T_a(W))$ with $W = (1, \cdots, 1)^T$ and $a = (1e4, 1e4, 1e-4, 1e-4) > 0$. The figure on the left hand side is the training loss of our PSI-SGD and SGD under an unbalanced initialization.

the order of $O(1/T)$ and $O(1/\sqrt{T})$ under the non-convex and smooth assumptions, where $T$ is the number of iterations. The results match the optimal one with fine-tuned learning rates [Ghadimi and Lan, 2013] under non-convex optimization problem. Besides that, the proposed algorithms are actually training with adaptive learning rate, which is decided by the local smoothness of loss function. The adaptive learning rate accelerates the training process as our proposed algorithms have improved constant dependence in the convergence rates, compared with their vanilla versions on the Euclidean space.

To empirically study the proposed algorithms, we compare PSI-SGD and PSI-SGDM with the other three baseline algorithms. They are respectively vanilla SGD, adaptive learning rate algorithm Adam [Diederik P. Kingma, 2015], and another manifold based algorithm SGDG [Cho and Lee, 2017]. The experiments are conducted on image classification task on datasets CIFAR and ImageNet [Krizhevsky and Hinton, 2009, Deng et al., 2009]. We empirical observe the proposed methods consistently improve the baseline algorithms in the sense of both convergence rate and generalization over various experimental settings. The observed accelerated convergence rate and better generalization ability of PSI-GD and PSI-SGD verify our theoretical justification that the two method are able to find local minima with less iterations.

## 2 RELATED WORKS

**Optimization on Manifold.** Absil et al. [2009] presents an introduction and summarization to this topic. The foundations and some recent theoretical results in this topic refers to [Absil and Gallivan, 2006, Liu et al., 2017, Zhang and Sra, 2016, Boumal et al., 2019]. Roughly speaking, optimization on manifold converts the constrained problem into an unconstrained problem on a specific manifold. **?**Lezcano-Casado

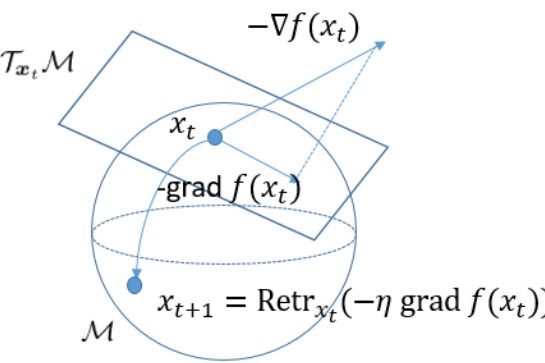

Figure 2: Gradient descent on Riemannian manifold.

and Martınez-Rubio [2019], Casado [2019], Li et al. [2019] leverage the technique to handle some explicit constraints to deep neural networks e.g., orthogonal constraints. [Cho and Lee, 2017, Huang et al., Badrinarayanan et al., 2015] fix the ambiguity of optimizing network with BN by manifold based algorithms. However, in contrast to this paper, some of the theoretical properties of their methods are missing, e.g., the convergence rate.

**PSI property.** Our method in this paper is build upon PSI property brought by BN. The PSI property properly defined in Section 4 are widely appears in modern deep neural networks. For example, the network with ReLU activation [Neyshabur et al., 2015] and the network with normalization layer, e.g., batch normalization [Ioffe and Szegedy, 2015], layer normalization [Ba et al.] and group normalization [Wu and He, 2018]. Arora et al. [2018], Wu et al. [2018] use a similar theoretical framework as ours to show the network with PSI property allows a more flexible choice of the learning rate, and thus easier to be trained. However, we design algorithm to obviate the optimization brought by PSI property, and show the proposed methods can have

improved convergence rate.

Due to the PSI property, Li and Arora [2019] uses exponentially increased learning rate for the network with BN to handle implicit effect brought by weight decay. In contrast to fixing the ambiguity of optimization by manifold based methods, Neyshabur et al. [2015], Meng et al. [2018], Zheng et al. [2019] re-parameterize the network by "path value" and optimizing on the path value space.

## 3 BACKGROUND

### 3.1 PROBLEM SET UP

In this paper, we consider the deep neural network with the following structure

$$f(\boldsymbol{x}, \boldsymbol{W}) = BN(\boldsymbol{W}_L \phi(BN(\boldsymbol{W}_{L-1} \phi(\cdots \phi(BN(\boldsymbol{W}_1 \boldsymbol{x})))))), \tag{1}$$

where $\phi(\cdot)$ is the non-linear activation, and $\boldsymbol{W}_l$ is the weight matrix of the $l$-th layer. $BN(\cdot)$ operator in the equation represents batch normalization [Ioffe and Szegedy, 2015] layer, which normalizes every hidden output $z = \boldsymbol{\theta}^T \boldsymbol{x}$ across a batch of training samples as

$$BN(\boldsymbol{z}) = \gamma \frac{\boldsymbol{z} - \mathbb{E}(\boldsymbol{z})}{\sqrt{\mathrm{Var}(\boldsymbol{z})}} + \beta = \gamma \frac{\boldsymbol{\theta}^T(\boldsymbol{x} - \mathbb{E}(\boldsymbol{x}))}{\sqrt{\boldsymbol{\theta}^T \mathrm{Var}_{\boldsymbol{x}} \boldsymbol{\theta}}} + \beta, \tag{2}$$

where $\gamma$ and $\beta$ are learnable affine parameters. The loss is

$$\mathcal{L}(\boldsymbol{W}, \boldsymbol{g}) = \frac{1}{n} \sum_{i=1}^{n} \ell(f(\boldsymbol{x}_i, \boldsymbol{W}), y_i), \tag{3}$$

across this paper, where $\{(\boldsymbol{x}_i, y_i)\}$ is the training set, $\ell(\boldsymbol{x}, y)$ represents the loss function, and $\boldsymbol{g}$ are scale-variant parameters i.e. $\gamma$ and $\beta$ in BN layer. Let $\boldsymbol{W} = (\boldsymbol{w}^{(1)} \cdots, \boldsymbol{w}^{(m)})$, the $\boldsymbol{w}^{(i)}$ is the weight vector of the $i$-th neuron. Then $\mathcal{L}(\boldsymbol{W}, \boldsymbol{g})$ is positively scale-invariant w.r.t. $\boldsymbol{W}$. It means $\mathcal{L}(\boldsymbol{W}, \boldsymbol{g}) = \mathcal{L}(T_{\boldsymbol{a}}(\boldsymbol{W}), \boldsymbol{g})$ for any $T_{\boldsymbol{a}}(\boldsymbol{W}) = (a_1 \boldsymbol{w}^{(1)}, \cdots, a_m \boldsymbol{w}^{(m)})$ with $\{a_i > 0\}$. In the right hand side of Figure 1, we illustrate such PSI property (the network is invariant for rescaling operator $T_{\boldsymbol{a}}(\cdot)$) via a two layer network, where the number between two circles (neurons) is the weight between them. Specially, let $\boldsymbol{V}$ be the normalized $\boldsymbol{W}$ which means $\boldsymbol{V} = (\boldsymbol{w}^{(1)}/\|\boldsymbol{w}^{(1)}\|, \cdots, \boldsymbol{w}^{(m)}/\|\boldsymbol{w}^{(m)}\|)$. Then we have $\mathcal{L}(\boldsymbol{V}, \boldsymbol{g}) = \mathcal{L}(\boldsymbol{W}, \boldsymbol{g})$.

It worth noting that our discussions below can be applied on the top of any functions with PSI property e.g., ReLU neural networks [Meng et al., 2018, Neyshabur et al., 2015]. However, we use NN with BN as a representation to illustrate our conclusions.

### 3.2 OPTIMIZATION ON MANIFOLD

We present a brief introduction to some definitions appeared in the optimization on manifold, more details are deferred

to Appendix A or [Absil et al., 2009]. A Matrix Manifold $\mathcal{M}$ is a subset of Euclidean space that for any $\boldsymbol{x} \in \mathcal{M}$, there exists a neighborhood $U_{\boldsymbol{x}}$ of $\boldsymbol{x}$, such that $U_{\boldsymbol{x}}$ is homeomorphic to a Euclidean space. It has been pointed out that the PSI parameters $\boldsymbol{W}$ can be defined on a matrix manifold, e.g., Grassmann manifold or Oblique manifold without losing representation capacity of model [Cho and Lee, 2017, Huang et al.]. For any $\boldsymbol{x} \in \mathcal{M}$, there is a tangent space $\mathcal{T}_{\boldsymbol{x}}\mathcal{M}$, such that for any $f(\cdot)$ defined on $\mathcal{M}$, its Riemannian gradient $\mathrm{grad} f(\boldsymbol{x}) \in \mathcal{T}_{\boldsymbol{x}}\mathcal{M}$ [1].

With a given retraction function $R_{\boldsymbol{x}}(\cdot) : \mathcal{T}_{\boldsymbol{x}}\mathcal{M} \to \mathcal{M}$, it was shown in [Boumal et al., 2019] that the gradient descent on manifold (refers to Figure 2)

$$\boldsymbol{x}_{t+1} = R_{\boldsymbol{x}_t}\left(-\eta \mathrm{grad} f(\boldsymbol{x}_t)\right) \tag{4}$$

has the same convergence rate with vanilla gradient descent in Euclidean space under some mild assumptions. Roughly speaking, the Riemannian gradient descent creates a series of iterates moving along the direction of $\mathrm{grad} f(\boldsymbol{x})$, while the moving is conducted via the pre-defined retraction function $R_{\boldsymbol{x}}(\cdot)$.

## 4 PSI MANIFOLD

The optimization ambiguity of PSI parameters states as follows. By chain rule,

$$\nabla_{\boldsymbol{w}^{(i)}} \mathcal{L}(\boldsymbol{W}, \boldsymbol{g}) = a_i \nabla_{a_i \boldsymbol{w}^{(i)}} \mathcal{L}(T_{\boldsymbol{a}}(\boldsymbol{W}), \boldsymbol{g}), \tag{5}$$

so the two positively scale-equivalent points $\boldsymbol{W}$ and $T_{\boldsymbol{a}}(\boldsymbol{W})$ with the same output converge to different local optima owing to the different gradients. In other words, gradient-based algorithms in Euclidean space can not guarantee two iterates started from $\boldsymbol{W}$ and $T_{\boldsymbol{a}}(\boldsymbol{W})$ positively scale equivalent with each other across the training. This makes the converged point becomes sensitive the scale of weights.

Unfortunately, the sensitivity sometimes makes model converge to a local minima with poor performance. For example, we train a simple two-layer neural network with BN on MNIST [LeCun et al., 1998], which is a image classification task. The network is trained with a unbalanced initialization where the unbalanced here means the weights in each layer are in different scales. As can be seen, in the left hand side of Figure 1, in contrast to our PSI-SGD (see Section 4.2), SGD performs very poorly under such initialization. However, the poor performance of SGD does adopted under balanced initialization [LeCun et al., 1998]. Thus the optimization ambiguity can break the success of training.

To obviate this, we construct a quotient manifold, i.e., PSI manifold. Constraining the model's parameters in such manifold can fix the training ambiguity brought by the positively scale-invariant property.

---

[1] According to [Absil et al., 2009], the Riemannian gradient $\mathrm{grad} f(\boldsymbol{x})$ sometimes is the projection of gradient $\nabla f(x)$ to the tangent space $\mathcal{T}_{\boldsymbol{x}}$.

## 4.1 CONSTRUCTION OF PSI MANIFOLD

A quotient manifold is induced by an equivalent relationship. We formally formulate the positively scale-equivalence to derive the PSI manifold with PSI parameters defined in it.

**Definition 1** (Positively Scale-Equivalent). *For a pair of $\boldsymbol{W}, \boldsymbol{W}'$, they are positively scale-equivalent with each other, if there exists a transformation $T_{\boldsymbol{a}}(\cdot)$ such that $\boldsymbol{W}' = T_{\boldsymbol{a}}(\boldsymbol{W})$. $\boldsymbol{W} \sim \boldsymbol{W}'$ denotes the equivalence.*

We can verify that the positively scale-equivalent "∼" is an equivalent relationship. The Proposition 3.4.6 in [Absil et al., 2009] implies the following proposition which constructs the PSI manifold.

**Proposition 1.** *For PSI parameters $\boldsymbol{W}$, the positively scale-equivalent relationship $\sim$ induces a quotient manifold $\mathcal{M} = \overline{\mathcal{M}}/\sim = \mathbb{R}^{d_1} \times \cdots \mathbb{R}^{d_m}/\sim$ named PSI manifold. Here $d_i$ is the dimension of $\boldsymbol{w}^{(i)}$.*

We formally defined the PSI manifold. Intuitively, all the parameters that are positively scale-equivalent with each other in the Euclidean space are viewed as the same element in the PSI manifold $\mathcal{M}$. Then, we can constrain the PSI parameters directly in the manifold without losing the representation ability of the model.

The PSI manifold is proposed to obviate the ambiguity of optimization brought by the PSI property. To this end, we can optimize the PSI parameters directly in the PSI manifold, since all the positively scale-equivalent parameters are the same point in it. But (4) implies that optimization on PSI manifold requires the formulation of Riemannian gradient and retraction function on it. The following proposition defines a Riemannian metric in the PSI manifold, which formulates the Riemannian gradient (See the definition in Appendix A).

**Proposition 2.** *For every $\boldsymbol{W} \in \mathcal{M}$ [2], the Riemannian metric $\langle \cdot, \cdot \rangle_{\boldsymbol{W}}$ in the PSI manifold can be defined as*

$$\langle \boldsymbol{\Xi}_1, \boldsymbol{\Xi}_2 \rangle_{\boldsymbol{W}} = \sum_{i=1}^{m} \frac{\langle \boldsymbol{\xi}_1^{(i)}, \boldsymbol{\xi}_2^{(i)} \rangle}{\|\boldsymbol{w}^{(i)}\|^2}, \tag{6}$$

*where $\boldsymbol{\Xi}_1, \boldsymbol{\Xi}_2 \in \mathcal{T}_{\boldsymbol{W}}\mathcal{M}$, with $\boldsymbol{\Xi}_k = (\boldsymbol{\xi}_k^{(1)}, \cdots \boldsymbol{\xi}_k^{(m)})$.*

The proof of this Proposition is in Appendix B.1. With the Riemannian metric, we can exactly compute the Riemannian gradient on the PSI manifold. According to Definition of Riemannian gradient in Appendix A, we have

$$\mathrm{grad}_{\boldsymbol{w}^{(i)}}\mathcal{L}(\boldsymbol{W}, \boldsymbol{g}) = \|\boldsymbol{w}^{(i)}\|^2 \nabla_{\boldsymbol{w}^{(i)}}\mathcal{L}(\boldsymbol{W}, \boldsymbol{g}) \tag{7}$$

for each $\boldsymbol{w}^{(i)}$. The retraction function $R_{\boldsymbol{W}}(\cdot)$ is required to obtain gradient-based algorithms on the PSI manifold.

---

[2] Please note that any $T_{\boldsymbol{a}}(\boldsymbol{W})$ and $\boldsymbol{W}$ are view as the same element in the PSI manifold $\mathcal{M}$. Here $\boldsymbol{W}$ is a representation of set $\{T_{\boldsymbol{a}}(\boldsymbol{W}) : a_i > 0\}$.

**Proposition 3.** *For every $\boldsymbol{W} \in \mathcal{M}, \boldsymbol{\Xi} \in \mathcal{T}_{\boldsymbol{W}}\mathcal{M}$, a retraction function $R_{\boldsymbol{W}}(\boldsymbol{\Xi})$ on the PSI manifold can be*

$$R_{\boldsymbol{W}}(\boldsymbol{\Xi}) = \boldsymbol{W} + \boldsymbol{\Xi}. \tag{8}$$

The proof of this proposition appears in Appendix C.1. Now we are ready to optimize PSI parameters in the PSI manifold directly, which obviates the ambiguity of training

## 4.2 OPTIMIZATION ON THE PSI MANIFOLD

In this subsection, we give the update rules of GD, SGD, and SGD with momentum on the PSI manifold, abbreviated as PSI-GD, PSI-SGD, and PSI-SGDM respectively. We show that all of these update rules on the PSI manifold generate a unified optimization path for parameters that positively scale-equivalent with each other.

Combining (4) and Proposition 2, 3, we have the following update rule of GD

$$\begin{aligned} \boldsymbol{w}_{t+1}^{(i)} &= R_{\boldsymbol{w}_t^{(i)}}\left(-\eta_{\boldsymbol{w}_t^{(i)}}\mathrm{grad}_{\boldsymbol{w}_t^{(i)}}\mathcal{L}(\boldsymbol{W}_t, \boldsymbol{g}_t)\right) \\ &= \boldsymbol{w}_t^{(i)} - \eta_{\boldsymbol{w}_t^{(i)}}\|\boldsymbol{w}_t^{(i)}\|^2 \nabla_{\boldsymbol{w}_t^{(i)}}\mathcal{L}(\boldsymbol{W}_t, \boldsymbol{g}_t), \end{aligned} \tag{9}$$

SGD

$$\begin{aligned} \boldsymbol{w}_{t+1}^{(i)} &= R_{\boldsymbol{w}_t^{(i)}}\left(-\eta_{\boldsymbol{w}_t^{(i)}}\mathrm{grad}_{\boldsymbol{w}_t^{(i)}}\hat{\mathcal{L}}(\boldsymbol{W}_t, \boldsymbol{g}_t)\right) \\ &= \boldsymbol{w}_t^{(i)} - \eta_{\boldsymbol{w}_t^{(i)}}\|\boldsymbol{w}_t^{(i)}\|^2 \mathcal{G}_{\boldsymbol{w}_t^{(i)}}(\boldsymbol{W}_t, \boldsymbol{g}_t), \end{aligned} \tag{10}$$

and SGD with momentum

$$\begin{aligned} \boldsymbol{u}_{t+1}^{(i)} &= R_{\rho\boldsymbol{u}_t^{(i)}}\left(-\eta_{\boldsymbol{w}_t^{(i)}}\mathrm{grad}_{\boldsymbol{w}_t^{(i)}}\hat{\mathcal{L}}(\boldsymbol{W}_t, \boldsymbol{g}_t)\right) \\ &= \rho\boldsymbol{u}_t^{(i)} - \eta_{\boldsymbol{w}_t^{(i)}}\|\boldsymbol{w}_t^{(i)}\|^2 \mathcal{G}_{\boldsymbol{w}_t^{(i)}}(\boldsymbol{W}_t, \boldsymbol{g}_t); \\ \boldsymbol{w}_{t+1}^{(i)} &= R_{\boldsymbol{w}_t^{(i)}}\left(\boldsymbol{u}_{t+1}^{(i)}\right) = \boldsymbol{w}_{t+1}^{(i)} + \boldsymbol{u}_{t+1}^{(i)}. \end{aligned} \tag{11}$$

for PSI parameters. Here

$$\begin{aligned} \mathrm{grad}_{\boldsymbol{w}_t^{(i)}}\hat{\mathcal{L}}(\boldsymbol{W}_t, \boldsymbol{g}_t) &= \|\boldsymbol{w}_t^{(i)}\|^2 \mathcal{G}_{\boldsymbol{w}_t^{(i)}}(\boldsymbol{W}_t, \boldsymbol{g}_t) \\ &= \|\boldsymbol{w}_t^{(i)}\|^2 \nabla_{\boldsymbol{w}_t^{(i)}} \frac{1}{S}\sum_{k=1}^{S}\nabla_{\boldsymbol{w}_t^{(i)}}\ell\left(f(\boldsymbol{x}_k, \boldsymbol{W}_t), y_k\right) \end{aligned} \tag{12}$$

for a batch of $\{(\boldsymbol{x}_1, y_1), \cdots, (\boldsymbol{x}_S, y_S)\}$ is an unbiased estimation to the Riemannian gradient. In addition, the update rules of non-scale-invariant parameters $\boldsymbol{g}$ follows the gradient-based algorithms in Euclidean space.

**Remark 1.** *The proposed PSI-GD is similar to the one in [Badrinarayanan et al., 2015]. However, they do not prove that the update rule is induced by a retraction function. Besides that, our theoretical characterization of its convergence rate is absent in their work.*

One can justify that the proposed algorithms on PSI manifold are essentially using adaptive learning rates decided by $\|\boldsymbol{w}_t^{(i)}\|^2$ to match the gradient scale brought by a variant of weight scale. The following proposition shows the well-posedness of the algorithms on PSI manifold.

**Algorithm 1** SGD with momentum on the PSI manifold (PSI-SGDM).

---

**Input**: Training steps $T$; batch size $S$; momentum parameters $\rho$; learning rate $\eta_{\boldsymbol{w}_t^{(i)}}$ and $\eta_{\boldsymbol{g}_t}$.

**for** $t = 0 \cdots T-1$ **do**

    Sampling a batch of data $\{(x_1, y_1), \cdots, (\boldsymbol{x}_S. y_S)\}$ from training set

    **for** $i = 1, \cdots, m$ **do**

      $\boldsymbol{u}_{t+1}^{(i)} =$
$\rho \boldsymbol{u}_t^{(i)} - \eta_{\boldsymbol{w}_t^{(i)}} \|\boldsymbol{w}_t^{(i)}\|^2 \frac{1}{S} \sum_{k=1}^S \nabla_{\boldsymbol{w}_t^{(i)}} \ell\left(f(\boldsymbol{x}_k, \boldsymbol{W}_t), y_k\right)$

      $\boldsymbol{w}_{t+1}^{(i)} = \boldsymbol{w}_{t+1}^{(i)} + \boldsymbol{u}_{t+1}^{(i)}$

    **end for**

    $\boldsymbol{g}_{t+1} = \boldsymbol{g}_t - \eta_{\boldsymbol{g}_t} \sum_{k=1}^S \nabla_{\boldsymbol{g}_t} \ell\left(f(\boldsymbol{x}_k, \boldsymbol{W}_t), y_k\right)$

**end for**

**return** $(\boldsymbol{W}_T, \boldsymbol{g}_T)$

---

**Theorem 1.** *For two positively scale-equivalent weights $\boldsymbol{W}_0$ and $T_{\boldsymbol{a}}(\boldsymbol{W}_0)$, let $\boldsymbol{W}_t$, $\hat{\boldsymbol{W}}_t$ be $t$-th iterate of PSI-SGDM respectively started from $\{\boldsymbol{W}_0, \boldsymbol{U}_0\}$ and $\{T_{\boldsymbol{a}}(\boldsymbol{W}_0), T_{\boldsymbol{a}}(\boldsymbol{U}_0)\}$. Then we have $\hat{\boldsymbol{W}}_t = T_{\boldsymbol{a}}(\boldsymbol{W}_t)$.*

This theorem can be easily generalized to PSI-GD and PSI-SGD; the proof of it is in Appendix C.1. The conclusion shows that the iterates generated by the update rules on the PSI manifold are equivalent with respect to applying $T_{\boldsymbol{a}}(\cdot)$. Hence, optimizing on the PSI manifold obviates the optimization ambiguity brought by the PSI property. The complete algorithm of PSI-SGDM refers to Algorithm 1. With $\rho = 0$, the PSI-SGDM degenerates to PSI-SGD.

**Remark 2.** *The update rule of SGD with momentum on manifold [Cho and Lee, 2017, Liu et al., 2017] involves the parallel transformation to make sure the $\boldsymbol{U}_{t+1}$ locates in the tangent space $\mathcal{T}_{\boldsymbol{W}_t}\mathcal{M}$. However, it requires the proposed retraction in Proposition 3 to be an exponential retraction map [Absil et al., 2009], which may fail in-practical. Hence, we heuristically propose the PSI-SGDM without verifying the retraction function. Even though, our method is well-defined to positively scale-transformation.*

## 5 OPTIMIZATION ON PSI MANIFOLD ACCELERATES TRAINING

In this section, we give the convergence rates of PSI-GD and PSI-SGD, and show that they accelerate training compared with vanilla GD and SGD on the Euclidean space.

### 5.1 CONVERGENCE RESULTS

We give the convergence rates of PSI-GD and PSI-SGD (Algorithm 1) in this subsection. For the PSI parameters, let $\boldsymbol{V}_t = (\boldsymbol{w}_t^{(1)}/\|\boldsymbol{w}_t^{(1)}\|, \cdots, \boldsymbol{w}_t^{(m)}/\|\boldsymbol{w}_t^{(m)}\|)$ be the normalized parameters, we impose the following assumptions to

characterize the smoothness of loss function.

$$
\begin{aligned}
\left\|\nabla^2_{\boldsymbol{v}^{(i)}\boldsymbol{v}^{(j)}} \mathcal{L}(\boldsymbol{V}, \boldsymbol{g})\right\|_2 &\le L_{ij}^{\boldsymbol{vv}}, \\
\left\|\nabla^2_{\boldsymbol{v}^{(i)}\boldsymbol{g}} \mathcal{L}(\boldsymbol{V}, \boldsymbol{g})\right\|_2 &\le L_i^{\boldsymbol{vg}}, \\
\left\|\nabla^2_{\boldsymbol{g}} \mathcal{L}(\boldsymbol{V}, \boldsymbol{g})\right\|_2 &\le L^{\boldsymbol{gg}}.
\end{aligned}
\tag{13}
$$

Here $\|\cdot\|_2$ is the spectral norm of matrix. For PSI-SGD, we further impose the bounded variance assumption:

$$
\begin{aligned}
\mathbb{E}\left[\left\|\mathcal{G}_{\boldsymbol{w}_t^{(i)}}(\boldsymbol{W}_t, \boldsymbol{g}_t) - \nabla_{\boldsymbol{w}_t^{(i)}} \mathcal{L}(\boldsymbol{W}_t, \boldsymbol{g}_t)\right\|^2\right] &\le \sigma^2; \\
\mathbb{E}\left[\left\|\mathcal{G}_{\boldsymbol{g}_t}(\boldsymbol{W}_t, \boldsymbol{g}_t) - \nabla_{\boldsymbol{w}_t^{(i)}} \mathcal{L}(\boldsymbol{W}_t, \boldsymbol{g}_t)\right\|^2\right] &\le \sigma^2.
\end{aligned}
\tag{14}
$$

**Remark 3.** *Due to (5), the upper bound (14) can not hold as $\|\boldsymbol{w}_i\|$ goes to zero. However, due to Lemma 1 below, the weights $\boldsymbol{w}_i$ obtained by GD or SGD will never goes to zero. Thus the bounded variance (14) is reasonable in this regime.*

Let $\mathcal{L}(\boldsymbol{W}^*, \boldsymbol{g}^*) = \inf_{\boldsymbol{W}, \boldsymbol{g}} \mathcal{L}(\boldsymbol{W}, \boldsymbol{g})$, the following theorems give the convergence rates of PSI-GD and PSI-SGD.

**Theorem 2.** *Let $\{\boldsymbol{W}_t, \boldsymbol{g}_t\}$ be the iterates of PSI-GD (9), then we have*

$$
\min_{0 \le t < T} \|\nabla_{\boldsymbol{V}_t, \boldsymbol{g}_t} \mathcal{L}(\boldsymbol{V}_t, \boldsymbol{g}_t)\|^2 \le \frac{2\tilde{L}(\mathcal{L}(\boldsymbol{W}_0, \boldsymbol{g}_0) - \mathcal{L}(\boldsymbol{W}^*, \boldsymbol{g}^*))}{T}
\tag{15}
$$

*by choosing $\eta_{\boldsymbol{w}_t}^{(i)} = 1/\tilde{L}_{\boldsymbol{v}^{(i)}}$ and $\eta_{\boldsymbol{g}_t} = 1/\tilde{L}_{\boldsymbol{g}}$. Here $\tilde{L} = \max\{\tilde{L}_{\boldsymbol{v}^{(1)}}, \cdots, \tilde{L}_{\boldsymbol{v}^{(m)}}, \tilde{L}_{\boldsymbol{g}}\}$, where $\tilde{L}_{\boldsymbol{v}^{(i)}} = L_i^{\boldsymbol{vg}} + \sum_{j=1}^m L_{ij}^{\boldsymbol{vv}}; \tilde{L}_{\boldsymbol{g}} = mL^{\boldsymbol{gg}} + \sum_{i=1}^m L_i^{\boldsymbol{vg}}$.*

**Theorem 3.** *Let $\{\boldsymbol{W}_t, \boldsymbol{g}_t\}$ be the iterates of PSI-SGD (10), then we have*

$$
\min_{0 \le t < T} \mathbb{E}\left[\|\nabla_{\boldsymbol{V}_t, \boldsymbol{g}_t} \mathcal{L}(\boldsymbol{V}_t, \boldsymbol{g}_t)\|^2\right] \le \frac{2\tilde{L}(\mathcal{L}(\boldsymbol{W}_0, \boldsymbol{g}_0) - \mathcal{L}(\boldsymbol{W}^*, \boldsymbol{g}^*))}{\sqrt{T}}
$$
$$
+ \frac{\sigma^2 \tilde{L}}{2\sqrt{T}} \sum_{i=1}^m \left(\frac{1}{\tilde{L}_{\boldsymbol{g}}} + \frac{1}{\tilde{L}_{\boldsymbol{v}^{(i)}}}\right),
\tag{16}
$$

*by choosing $\eta_{\boldsymbol{w}_t^{(i)}} = \frac{1}{\tilde{L}_{\boldsymbol{v}^{(i)}} \sqrt{T}}$ and $\eta_{\boldsymbol{g}_t} = \frac{1}{\tilde{L}_{\boldsymbol{g}} \sqrt{T}}$, where $\tilde{L}, \tilde{L}_{\boldsymbol{v}^{(i)}}$ and $\tilde{L}_{\boldsymbol{g}}$ are defined in Theorem 2.*

The theorems are respectively proved in Appendix D.3 and D.4. The convergence rates of PSI-GD and PSI-SGD are respectively $O(1/T)$ and $O(1/\sqrt{T})$. The convergence rates of PSI-SGDM can be also obtained by combining the technique in [Yang et al., 2016] and our proof of Theorem 3.

One may note the convergence rate is computed on the gradient w.r.t. normalized parameters $\boldsymbol{V}_t$. But (5) implies

$$
\begin{aligned}
\|\nabla_{\boldsymbol{W}_t, \boldsymbol{g}_t} \mathcal{L}(\boldsymbol{W}_t, \boldsymbol{g}_t)\|^2 &= \sum_{i=1}^m \frac{1}{\|\boldsymbol{w}_t^{(i)}\|^2} \left\|\nabla_{\boldsymbol{v}_t^{(i)}, \boldsymbol{g}_t} \mathcal{L}(\boldsymbol{V}_t, \boldsymbol{g}_t)\right\|^2 \\
&\quad + \|\nabla_{\boldsymbol{g}_t} \mathcal{L}(\boldsymbol{V}_t, \boldsymbol{g}_t)\|^2 \\
&\le \|\nabla_{\boldsymbol{V}_t, \boldsymbol{g}_t} \mathcal{L}(\boldsymbol{V}_t, \boldsymbol{g}_t)\|^2,
\end{aligned}
\tag{17}
$$

if $\|\boldsymbol{w}_t^{(i)}\|^2 \geq 1$ for $1 \leq i \leq m$. Thus we can get the corresponded convergence rates of PSI-GD and PSI-SGD. They match the optimal results with well tuned learning rates in the Euclidean space [Ghadimi and Lan, 2013]. The following lemma shows that the $\|\boldsymbol{w}_t^{(i)}\|^2$ keeps increasing across training for gradient-based algorithms.

**Lemma 1** (Lemma 2.4 in [Arora et al., 2018]). *For any PSI weight $\boldsymbol{w}^{(i)}$, $\boldsymbol{w}^{(i)}$ and $\nabla_{\boldsymbol{w}^{(i)}}\ell\left(f(\boldsymbol{x}_k, \boldsymbol{W}), y_k\right)$ are perpendicular for any $(\boldsymbol{x}_k, y_k)$. On the other hand*

$$
\begin{aligned}
&\left\|\boldsymbol{w}^{(i)} + \eta_{\boldsymbol{w}^{(i)}} \nabla_{\boldsymbol{w}^{(i)}}\ell\left(f(\boldsymbol{x}_k, \boldsymbol{W}), y_k\right)\right\|^2 = \left\|\boldsymbol{w}^{(i)}\right\|^2 \\
&+ \eta_{\boldsymbol{w}^{(i)}}^2 \left\|\nabla_{\boldsymbol{w}^{(i)}}\ell\left(f(\boldsymbol{x}_k, \boldsymbol{W}), y_k\right)\right\|^2 .
\end{aligned}
\tag{18}
$$

Thus, $\|\boldsymbol{w}_t^{(i)}\|^2 \geq 1$ holds, if $\|\boldsymbol{w}_0^{(i)}\|^2 \geq 1$. Since the network is usually initialized by $\boldsymbol{w}^{(i)} \sim \mathcal{N}(0, 2/\sqrt{d_i})$ [He et al., 2015], $\|\boldsymbol{w}_0^{(i)}\|^2 \approx 2$ with high probability. Hence, we can conclude $\left\|\nabla_{\boldsymbol{W}_t, \boldsymbol{g}_t}\mathcal{L}(\boldsymbol{W}_t, \boldsymbol{g}_t)\right\|^2 \leq \left\|\nabla_{\boldsymbol{V}_t, \boldsymbol{g}_t}\mathcal{L}(\boldsymbol{V}_t, \boldsymbol{g}_t)\right\|^2$. We will show the inequality and the increasing $\|\boldsymbol{w}_t^{(i)}\|^2$ explain the acceleration of the algorithms on PSI manifold.

## 5.2 WHY OPTIMIZATION ON THE PSI MANIFOLD ACCELERATES TRAINING

Next, we show PSI-GD and PSI-SGD accelerate training compared with the algorithms on Euclidean space.

Generally, PSI parameters move towards a smoother region (smaller gradient Lipschitz constant) across training, since the gradient Lipschitz constant is in inverse ratio to the weight scale which keeps increasing according to Lemma 1. As smoother region allows a larger learning rate, gradually increasing the learning rate accelerates training. Fortunately, PSI-GD and PSI-SGD happen to be vanilla GD and SGD with adaptively increased learning rates according to (9), (10) and (18).

To begin with, we verify the convergence rates of vanilla GD and SGD. We assume a global smoothness for $\mathcal{L}(\boldsymbol{W}, \boldsymbol{g})$,

$$
\begin{aligned}
\left\|\nabla^2_{\boldsymbol{w}^{(i)}\boldsymbol{w}^{(j)}}\mathcal{L}(\boldsymbol{W}, \boldsymbol{g})\right\|_2 &\leq L_{ij}^{\boldsymbol{ww}}, \\
\left\|\nabla^2_{\boldsymbol{w}^{(i)}\boldsymbol{g}}\mathcal{L}(\boldsymbol{W}, \boldsymbol{g})\right\|_2 &\leq L_{i}^{\boldsymbol{wg}}, \\
\left\|\nabla^2_{\boldsymbol{g}}\mathcal{L}(\boldsymbol{W}, \boldsymbol{g})\right\|_2 &\leq L^{\boldsymbol{gg}}.
\end{aligned}
\tag{19}
$$

The assumption is stronger than (13) since (13) only involves the normalized parameters, thus $L_{ij}^{\boldsymbol{vv}} \leq L_{ij}^{\boldsymbol{ww}}; L_i^{\boldsymbol{vg}} \leq L_i^{\boldsymbol{wg}}$. We consider the update rule of GD

$$
\begin{aligned}
\boldsymbol{w}_{t+1}^{(i)} &= \boldsymbol{w}_t^{(i)} - \eta_{\boldsymbol{w}_t^{(i)}} \nabla_{\boldsymbol{w}_t^{(i)}}\mathcal{L}(\boldsymbol{W}_t, \boldsymbol{g}_t); \\
\boldsymbol{g}_{t+1} &= \boldsymbol{g}_t - \eta_{\boldsymbol{g}_t} \nabla_{\boldsymbol{g}_t}\mathcal{L}(\boldsymbol{W}_t, \boldsymbol{g}_t),
\end{aligned}
\tag{20}
$$

and SGD

$$
\begin{aligned}
\boldsymbol{w}_{t+1}^{(i)} &= \boldsymbol{w}_t^{(i)} - \eta_{\boldsymbol{w}_t^{(i)}} \mathcal{G}_{\boldsymbol{w}_t^{(i)}}(\boldsymbol{W}_t, \boldsymbol{g}_t); \\
\boldsymbol{g}_{t+1} &= \boldsymbol{g}_t - \eta_{\boldsymbol{g}_t} \mathcal{G}_{\boldsymbol{g}_t}(\boldsymbol{W}_t, \boldsymbol{g}_t),
\end{aligned}
\tag{21}
$$

where $\mathcal{G}_{\boldsymbol{w}_t^{(i)}}(\boldsymbol{W}_t, \boldsymbol{g}_t)$ and $\mathcal{G}_{\boldsymbol{g}_t}(\boldsymbol{W}_t, \boldsymbol{g}_t)$ are respectively unbiased estimations of $\nabla_{\boldsymbol{w}_t^{(i)}}\mathcal{L}(\boldsymbol{W}_t, \boldsymbol{g}_t)$ and $\nabla_{\boldsymbol{g}_t}\mathcal{L}(\boldsymbol{W}_t, \boldsymbol{g}_t)$ defined in (12). For SGD, we assume the two estimators satisfy the bounded variance assumption (14).

The following two Theorems characterize the convergence rate of GD and SGD.

**Theorem 4.** *Let $\{\boldsymbol{W}_t, \boldsymbol{g}_t\}$ updated by GD (20), by choosing $\eta_{\boldsymbol{w}_t}^{(i)} = 1/\tilde{L}_{\boldsymbol{w}^{(i)}}$ and $\eta_{\boldsymbol{g}_t} = 1/\tilde{L}_{\boldsymbol{g}}$,*

$$
\min_{0 \leq t < T} \|\nabla_{\boldsymbol{W}_t, \boldsymbol{g}_t}\mathcal{L}(\boldsymbol{W}_t, \boldsymbol{g}_t)\|^2 \leq \frac{2\tilde{L}(\mathcal{L}(\boldsymbol{W}_0, \boldsymbol{g}_0) - \mathcal{L}(\boldsymbol{W}^*, \boldsymbol{g}^*))}{T}.
\tag{22}
$$

*Here $\tilde{L} = \max\{\tilde{L}_{\boldsymbol{w}^{(1)}}, \cdots, \tilde{L}_{\boldsymbol{w}^{(m)}}, \tilde{L}_{\boldsymbol{g}}\}$, where $\tilde{L}_{\boldsymbol{w}^{(i)}} = L_i^{\boldsymbol{wg}} + \sum_{j=1}^m L_{ij}^{\boldsymbol{ww}}; \tilde{L}_{\boldsymbol{g}} = mL^{\boldsymbol{gg}} + \sum_{i=1}^m L_i^{\boldsymbol{wg}}$.*

**Theorem 5.** *Let $\{\boldsymbol{W}_t, \boldsymbol{g}_t\}$ updated by SGD (21), then*

$$
\begin{aligned}
\min_{0 \leq t < T} \mathbb{E}\big[\|\nabla_{\boldsymbol{W}_t, \boldsymbol{g}_t}\mathcal{L}(\boldsymbol{W}_t, \boldsymbol{g}_t)\|^2\big] \leq & \frac{2\tilde{L}(\mathcal{L}(\boldsymbol{W}_0, \boldsymbol{g}_0) - \mathcal{L}(\boldsymbol{W}^*, \boldsymbol{g}^*))}{\sqrt{T}} \\
& + \frac{\sigma^2 \tilde{L}}{2\sqrt{T}} \sum_{i=1}^m \left(\frac{1}{\tilde{L}_{\boldsymbol{g}}} + \frac{1}{\tilde{L}_{\boldsymbol{w}^{(i)}}}\right),
\end{aligned}
\tag{23}
$$

*by choosing $\eta_{\boldsymbol{w}_t^{(i)}} = \frac{1}{\tilde{L}_{\boldsymbol{w}^{(i)}}\sqrt{T}}$ and $\eta_{\boldsymbol{g}_t} = \frac{1}{\tilde{L}_{\boldsymbol{g}}\sqrt{T}}$, where $\tilde{L}, \tilde{L}_{\boldsymbol{w}^{(i)}}$ and $\tilde{L}_{\boldsymbol{g}}$ are defined in Theorem 4.*

The two convergence rates of $\|\nabla_{\boldsymbol{W}, \boldsymbol{g}}\mathcal{L}(\boldsymbol{W}, \boldsymbol{g})\|$ match the optimal results with well tuned learning rates [Ghadimi and Lan, 2013]. Since $L_{ij}^{\boldsymbol{vv}} \leq L_{ij}^{\boldsymbol{ww}}; L_i^{\boldsymbol{vg}} \leq L_i^{\boldsymbol{wg}}$, the right hand side in (22) is smaller than the one in (15). To see the results for SGD, under (19), the upper bound in (16) can be replaced by the smaller one of the upper bounds in (16) and (23). Thus the convergence rate of PSI-SGD is also sharper than the one of SGD.

On the other hand, (17) and the discussion in the above section implies $\|\nabla_{\boldsymbol{W}_t, \boldsymbol{g}_t}\mathcal{L}(\boldsymbol{W}_t, \boldsymbol{g}_t)\|^2 \leq \|\nabla_{\boldsymbol{V}_t, \boldsymbol{g}_t}\mathcal{L}(\boldsymbol{V}_t, \boldsymbol{g}_t)\|^2$. Thus, from (17), one can verify the proposed algorithms accelerate training in a factor $\|\boldsymbol{w}_t^{(i)}\|^2 \geq 1$ compared with the versions on Euclidean space. In addition, $\|\boldsymbol{w}_t^{(i)}\|^2$ keeps increasing to a bounded constant across training, thus the acceleration is more significant after a number of iterations. A detailed discussion to $\|\boldsymbol{w}_t^{(i)}\|^2$ is in Appendix E.

The proofs of the two theorems are delegated to Appendix D.1 and D.2. In the proof, we see that scaling the learning rate with $1/\tilde{L}_{\boldsymbol{w}^{(i)}}$ and $1/\tilde{L}_{\boldsymbol{g}}$ are respectively the optimal schedule of $\eta_{\boldsymbol{w}_t^{(i)}}$ and $\eta_{\boldsymbol{g}_t}$. Since smaller $\tilde{L}_{\boldsymbol{w}^{(i)}}$ and $\tilde{L}_{\boldsymbol{g}}$ corresponds with a smoother loss landscapes, it explains that $\mathcal{L}(\boldsymbol{W}, \boldsymbol{g})$ allows a larger learning rate in a smoother region to get the optimal convergence rate.

Now we are ready to illustrate the reason of PSI-GD and PSI-SGD's acceleration. Due to $\mathcal{L}(\boldsymbol{W}, \boldsymbol{g}) = \mathcal{L}(T_{\boldsymbol{a}}(\boldsymbol{W}), \boldsymbol{g})$,

$$
\begin{aligned}
\nabla^2_{\boldsymbol{w}^{(i)}\boldsymbol{w}^{(j)}}\mathcal{L}(\boldsymbol{W}, \boldsymbol{g}) &= a_i a_j \nabla^2_{a_i\boldsymbol{w}^{(i)} a_j\boldsymbol{w}^{(j)}}\mathcal{L}(T_{\boldsymbol{a}}(\boldsymbol{W}), \boldsymbol{g}); \\
\nabla^2_{\boldsymbol{w}^{(i)}\boldsymbol{g}}\mathcal{L}(\boldsymbol{W}, \boldsymbol{g}) &= a_i \nabla^2_{a_i\boldsymbol{w}^{(i)}\boldsymbol{g}}\mathcal{L}(T_{\boldsymbol{a}}(\boldsymbol{W}), \boldsymbol{g}).
\end{aligned}
\tag{24}
$$

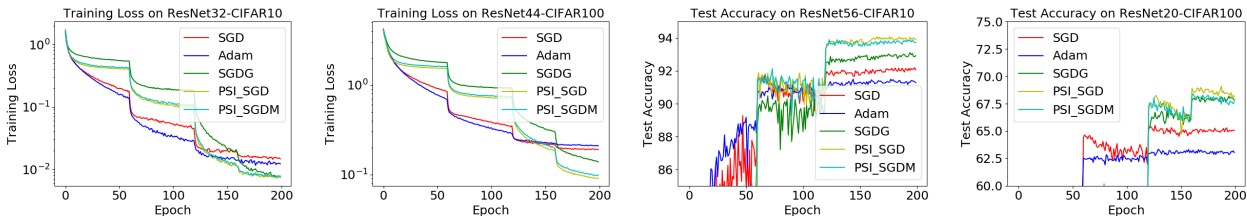

Figure 3: Results of ResNet trained over various Algorithms on `CIFAR10` and `CIFAR100`.

Table 1: Performance of ResNet on various datasets. The ResNet for `CIFAR` and `ImageNet` are respectively the structure for the corresponded dataset, more details about the structure refers to [He et al., 2016]. The results are the average over five (resp. three) independent runs for CIFAR (resp. ImageNet) with standard deviation reported.

| Dataset | CIFAR-10 | | | | |
|---|---|---|---|---|---|
| Algorithm | SGD | Adam | SGDG | PSI-SGD | PSI-SGDM |
| ResNet20 | 91.14(±0.10) | 89.90(±0.17) | 91.38(±0.21) | 92.25(±0.03) | **92.41**(±0.12) |
| ResNet32 | 91.32(±0.23) | 90.40(±0.32) | 92.48(±0.35) | **93.56**(±0.08) | 93.30(±0.14) |
| ResNet44 | 91.95(±0.14) | 91.02(±0.18) | 92.99(±0.12) | **93.90**(±0.16) | 93.39(±0.11) |
| ResNet56 | 92.19(±0.21) | 91.49(±0.12) | 93.11(±0.16) | **94.08**(±0.15) | 93.90(±0.16) |
| Dataset | CIFAR-100 | | | | |
| ResNet20 | 65.53(±0.51) | 63.35(±0.06) | 68.19(±0.24) | **69.11**(±0.07) | 68.41(±0.21) |
| ResNet32 | 67.41(±0.21) | 64.43(±0.11) | 70.13(±0.28) | **70.81**(±0.08) | 70.14(±0.13) |
| ResNet44 | 67.72(±0.31) | 65.03(±0.38) | 71.32(±0.12) | 71.94(±0.21) | **72.03**(±0.14) |
| ResNet56 | 68.02(±0.26) | 65.77(±0.13) | 71.46(±0.06) | **72.88**(±0.24) | 72.40(±0.16) |
| Dataset | ImageNet | | | | |
| ResNet18 | 67.72(±0.12) | 68.16(±0.13) | 68.97(±0.05) | **70.38**(±0.06) | 69.42(±0.06) |
| ResNet34 | 71.30(±0.08) | 70.68(±0.05) | 72.40(±0.12) | **73.31**(±0.08) | 72.88(±0.12) |
| ResNet50 | 73.46(±0.14) | 74.00(±0.16) | 74.67(±0.14) | 74.67(±0.12) | **75.11**(±0.05) |

The fact shows that the smoothness of PSI parameters increasing with its scale. As larger $\boldsymbol{a}$ in the right hand side of (24) results in smaller $\|\nabla^2_{a_i \boldsymbol{w}^{(i)} a_j \boldsymbol{w}^{(j)}} \mathcal{L}(T_{\boldsymbol{a}}(\boldsymbol{W}), \boldsymbol{g})\|_2$ and $\|\nabla^2_{a_i \boldsymbol{w}^{(i)} \boldsymbol{g}} \mathcal{L}(T_{\boldsymbol{a}}(\boldsymbol{W}), \boldsymbol{g})\|_2$. Due to Lemma 1, the increasing $\|\boldsymbol{w}_t^{(i)}\|$ implies the loss landscape of PSI parameters becomes smoother across training. Thus, gradually increasing the learning rate to update PSI parameters can accelerate training. The proposed PSI-GD and PSI-SGD are proven to be vanilla GD and SGD with the increasing learning rate that is proportional to $\|\boldsymbol{w}_t^{(i)}\|^2$, which interprets the acceleration.

Finally, (24) shows that our algorithms are optimal in the manner of leveraging the smoothness to accelerate training. The intuition is that the proposed methods have a more accurate estimation of local smoothness across training. We point out that $L_{ij}^{\boldsymbol{vv}} \leq L_{ij}^{\boldsymbol{ww}}; L_i^{\boldsymbol{vg}} \leq L_i^{\boldsymbol{wg}}$ in (19) also gives a sharper convergence rate of PSI-GD and PSI-SGD.

## 6 EXPERIMENTS

### 6.1 IMPROVED CONVERGENCE RATE

We use a toy example to verify that our algorithms have improved convergence rate compared with other methods.

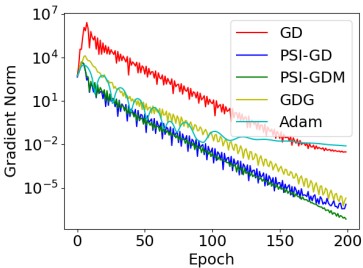

Figure 4: Convergence rates of gradient norm.

**Data.** We sample 1000 training samples $\{\boldsymbol{x}_i\}$ from 10-dimensional normal distribution with its label $y_i = \mu^\top \boldsymbol{x}_i + \epsilon_i$ for $\epsilon_i \sim \mathcal{N}(0, 1)$.

**Setup.** We use a toy neural network $f(\boldsymbol{x}) = \boldsymbol{w}_1^\top \phi(W_2 \boldsymbol{x})$ with $\boldsymbol{w}_1 \in \mathbb{R}^{100}$, $W_2 \in \mathbb{R}^{100 \times 10}$, and the activation function $\phi(\cdot)$ is ELU function. The training loss is MES loss. We compare the convergence rates in terms of gradient norm of the proposed algorithms PSI-SGD and PSI-SGDM with three baselines algorithms. The benchmark optimization algorithm SGD with momentum (abbrev. SGD); adaptive learning rate algorithm Adam [Diederik P. Kingma,

2015]; manifold based algorithm SGDG [Cho and Lee, 2017] which is applied to the network with BN.

**Main Results.** The convergence rates of gradient norm are in Figure 4. As can be seen, the three manifold based methods PSI-SGD, PSI-SGDM, SGDG exhibit improved convergence rates compared other methods. However, we still observe that PSI-SGD and PSI-SGDM are slightly better than SGDG. More important is that in contrast to SGDG, our algorithms have proved convergence rate. The toy example verifies that our algorithms have improved convergence rates compared with other methods.

## 6.2 EXPERIMENTS ON REAL-WORLD DATASET

In this section, we empirically study the proposed algorithms PSI-SGD and PSI-SGDM on real-world dataset.

**Data.** We consider the image classification task on three benchmark datasets. CIFAR10 and CIFAR100 [Krizhevsky and Hinton, 2009] are respectively colorful images with 50K training samples and 10K validation samples from 10 and 100 categories. ImageNet [Deng et al., 2009] are colorful images with 1M+ training samples from 1K object classes.

**Setup.** The model is a unified structure ResNet with various structures. As in the above section, we compare our methods with SGD, Adam, and SGDG. For PSI-SGD or PSI-SGDM, the PSI parameters are updated by the two algorithms, while the other parameters are updated by SGD.

We do not use the regularizer to the PSI weights e.g., $l_2$-regularizer in the loss function since it breaks the PSI property of PSI weights. More experiments conducted with regularizer are in Appendix F.

For CIFAR we conduct 200 epochs of training for each algorithm. The learning rate starts from 0.1 and decays by a factor 0.2 at epochs 60, 120, and 160. For ImageNet, the training is conducted for 100 epochs, and the learning rate starts from 0.1 and decays by a factor 0.2 at epochs 30, 60, and 90. For the hyperparameters of baseline methods, we grid search the learning rates in the range of {0.01, 0.1, 1.0} and {0.0001, 0.001, 0.01} respectively for SGD and Adam, and the hyperparameters of SGDG follow the one of [Cho and Lee, 2017]. The other hyperparameters of all these methods are summarized in Appendix F.2.

It worth noting that the PSI weights updated by PSI-SGD and PSI-SGDM may overflow after quite a number of iterations. Hence, we normalize the PSI weights $\boldsymbol{w}^{(i)}$ if $\|\boldsymbol{w}^{(i)}\|$ is larger than 10000 during training. This operation will not change the output of the model due to the PSI property.

**Main Results.** We report the test accuracy of each method. The results refers to Table 1 and Figure 3. We have the fol-

lowing observations and conclusions from the experimental results.

1. In Table 1, the model trained by manifold based methods i.e., SGDG, PSI-SGD and, PSI-SGDM generalize better in most cases. this is due to the manifold based algorithms can obviate quite a lot of local minima with poor generalization, since they have a unified optimization path for the parameters equivalent to each other.

   Besides that, the proposed PSI-SGD and PSI-SGDM are significantly better than SGDG, and the performances of PSI-SGD and PSI-SGDM are comparable. We speculate this is due to the sharper convergence rate of PSI-SGD and PSI-SGDM allow them to find local minima in less number of iterations.

2. Figure 3 show that the PSI-SGD and PSI-SGDM converge faster than the three baselines after a certain number of iterations i.e., after 120 epochs of update. This justifies our theoretical results in Section 5, since the acceleration is linearly with $\|\boldsymbol{w}_t^{(i)}\|^2$ which is large after a while of training (Lemma 1).

   We present the results of training error instead of gradient norm here because evaluating the gradient norm requires implementing back propagations on all training data which brings a great extra computational effort, especially for large scale dataset e.g., ImageNet.

We have one remark about the efficiency of our methods. They are simple and efficient compared with SGDG, since SGDG involves the operators like trigonometric functions in the update rule. For example, the elapsed time of ResNet56 with respect to one-epoch training of CIFAR10 under SGD, Adam, SGDG, PSI-SGD, and PSI-SGDM are respectively 16.98, 18.1, 26.6, 18.2, and 18.5 seconds. All the experiments are conducted on a server with single NVIDIA V100 GPU.

## 7 CONCLUSION

In this paper, we fix the optimization ambiguity brought by the PSI property of the network with BN. Our scenario is built upon optimization on manifold by constructing a specific manifold and optimizing the PSI weights of the network with BN in it. The developed gradient-based algorithms on PSI manifold are shown to have a well-defined optimization path with respect to positively equivalent rescaling.

We also give the convergence rates of the proposed methods. Besides that, we theoretically justify that PSI-GD and PSI-SGD accelerate training by a clever schedule of adaptive learning rate.

Finally, we conduct various experiments to show that the proposed methods have better performance in the generalization and efficiency compared with the other three baselines.

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
