# OpenReview forum: "Accelerating Training of Batch Normalization: A Manifold Perspective"
_auai.org/UAI/2022/Conference — UAI 2022 Poster_

### Official Review · Reviewer_zktB · 2022-04-03

**Q2(1) Originality/Novelty:** 3
**Q2(2) Significance/Impact:** 3
**Q2(3) Correctness/Technical Quality:** 3
**Q2(6) Clarity Of Writing:** 3
**Q6 Overall Score:** 6
**Q8 Confidence In Your Score:** 3

**Q1 Summary And Contributions:**

This paper studies acceleration of Batchnorm models by using a manifold gradient descent training method. They first propose the PSI manifold, which captures invariance of weights under positive rescalings in BatchNorm. They then propose GD/SGD/momentum algorithms  for optimization on this manifold, with convergence rate bounds for these methods. Empirical verifications of these algorithms show that they can improve empirical performance for ResNets trained on CIFAR.

**Q2 Assessment Of The Paper:**

More detailed information regarding each of these aspects is given below:

**Q2(4) Quality Of Experiments (Optional):**

3: Good: The experimental evaluation is adequate, and the results convincingly support the main claims.

**Q2(5) Reproducibility:**

2: Fair: Key resources (e.g., proofs, code, data) are unavailable but key details (e.g., proof sketches, experimental setup) are sufficiently well-described for an expert to confidently reproduce the main results.

**Q3 Main Strengths:**

The paper proposes a simple, but empirically effective algorithm for taking advantage of the scale invariance of BatchNorm/other normalization layers for accelerating optimization. While the formulation of the algorithm is quite simple and intuitive (and perhaps did not fully need the manifold optimization machinery to motivate it), the empirical improvements demonstrated in Section 6 are quite interesting.

**Q4 Main Weakness:**

The theoretical comparison in Section 5 between PSI-(S)GD and (S)GD seems to be somewhat limited by the fact that the comparison is of *upper bounds*. It would be more impactful to compare against a lower bound on regular GD. Regarding empirical evaluation, it would have been nice to see more settings in the main body of the paper (Section 6), e.g. different architecture sizes and normalization layers (Layernorm v.s. BatchNorm)? Finally, the results in Section 6 are obtained without $\ell_2$ regularization, which brings the baseline further away from practical settings. In the Appendix, it's shown that the gains of PSI-SGD are reduced when there is regularization, which makes it unclear whether PSI-SGD is helpful in all settings.

**Q5 Detailed Comments To The Authors:**

-- How is the last layer handled in the implementation of PSI-SGD? In most implementations, the network is not scale invariant w.r.t. this layer.
-- It's interesting that PSI-SGD seems to perform worse with momentum, whereas momentum typically seems to be helpful when training deep nets with standard SGD. Do the authors have any insights into this?
-- It appears to me that the algorithm could be motivated in a more simple manner, which is simply that scaling the update by $\|w\|_2^2$ makes it so that the ratio between $\|w\|_2$ and the norm of the update doesn't depend on $\|w\|_2$. Does the manifold gradient descent perspective offer any other benefit beyond this?

**Q7 Justification For Your Score:**

I think the algorithm proposed in this paper is quite interesting, and the experiments are also quite promising. However, there is still some unclarity about the empirical impacts in various settings, which prevents me from giving a higher score.

**Q9 Complying With Reviewing Instructions:**

1: Yes.

---

### Official Review · Reviewer_XpW5 · 2022-04-10

**Q2(1) Originality/Novelty:** 4
**Q2(2) Significance/Impact:** 3
**Q2(3) Correctness/Technical Quality:** 3
**Q2(6) Clarity Of Writing:** 1
**Q6 Overall Score:** 6
**Q8 Confidence In Your Score:** 3

**Q1 Summary And Contributions:**

While networks with batch normalization are invariant to positively re-scaling their weights, first-order optimization methods yield different outputs for initially equivalent weight matrices.

The authors propose to solve this inconsistency by proposing an equivariant optimization algorithm that yields equivalent solutions to equivalent networks. This algorithm does manifold gradient descent, and the authors provide different variants of existing algorithms with theoretical guarantees.

**Q2 Assessment Of The Paper:**

More detailed information regarding each of these aspects is given below:

**Q2(4) Quality Of Experiments (Optional):**

3: Good: The experimental evaluation is adequate, and the results convincingly support the main claims.

**Q2(5) Reproducibility:**

2: Fair: Key resources (e.g., proofs, code, data) are unavailable but key details (e.g., proof sketches, experimental setup) are sufficiently well-described for an expert to confidently reproduce the main results.

**Q3 Main Strengths:**

- The paper is groundbreaking to me. The problem itself is really interesting and the paper made me realize of its existence, and the simplicty of the proposed solution after that much math is surprising and elegant.
- Most of the maths can be easily followed in the main paper.
- Theoretical guarantees are provided, and the experiments show the advantage of the proposed method.

**Q4 Main Weakness:**

- 4.1 *Presentation.* The writing leaves a lot to be desired, and plenty of typos and grammatical errors make the paper really hard to read. Besides, figures are not explained and are too small at times, there are missing references, and plenty of empty space that could be better utilized.
- 4.2 Problem motivation is rather weak (see below).
- 4.3 All experimentation is performed on image datasets (CIFAR and ImageNet), and a single type of neural network (ResNets). Moreover, the impact of the proposed method is reduced when one realize that a regularized network obtains almost equivalent results to the proposed method (see Table 2 vs. Table 1).
- 4.4 Experiments  are performed on a single run (so no standard deviation reported), making it really hard to draw any conclusion (specially for Table 2).
- 4.5 Background, and specially SGDG, is almost not explained. Making it hard to understand the differences between the proposed method and SGDG.

**Q5 Detailed Comments To The Authors:**

- 5.1 I would appreciate more emphasis on the problem motivation. Either showing empirically the problem of this inconsistency between equivalent matrices, or showing the maths.
- 5.2 Remark 1 is a bit concerning. I would appreciate if the authors extend remark 1 and fully explain why not verifying that it is a retraction it not a problem.
- 5.3 Equation 17 confuses me a bit. Does it mean that the convergence rate is dependent on the chosen representation? Can this be exploited?

**Q7 Justification For Your Score:**

I love the idea of the paper and the simplicity of the proposed solution, but the presentation of the paper needs major revision.

At some point in my review, I stopped annotating writing issues, and I realized I could hardly focus on evaluating the technical problems of the work. Presentation issues are major enough to shadow the rest of the paper.

==

After the discussion, the authors have addressed my technical concerns, and I trust they will put enough effort to improve the presentation.

**Q9 Complying With Reviewing Instructions:**

1: Yes.

---

### Official Review · Reviewer_Lthz · 2022-04-12

**Q2(1) Originality/Novelty:** 2
**Q2(2) Significance/Impact:** 2
**Q2(3) Correctness/Technical Quality:** 3
**Q2(6) Clarity Of Writing:** 3
**Q6 Overall Score:** 6
**Q8 Confidence In Your Score:** 4

**Q1 Summary And Contributions:**

The paper considers the batch normalization as the optimization of positive-scale invariant (PSI) manifold. This is a Riemannian manifold, so specialized methods are proposed: gradient descent, stochastic gradient descent and SGD with momentum, all on the manifold. The idea closely follows prior work of Cho et. al in 2017, however it provides theoretical analysis and numerical experiments which confirm that the new methods outperform the previous ones.

**Q2 Assessment Of The Paper:**

More detailed information regarding each of these aspects is given below:

**Q2(4) Quality Of Experiments (Optional):**

3: Good: The experimental evaluation is adequate, and the results convincingly support the main claims.

**Q2(5) Reproducibility:**

4: Excellent: Key resources (e.g., proofs, code, data) are available and key details (e.g., proof sketches, experimental setup) are comprehensively described for competent researchers to confidently and easily reproduce the main results.

**Q3 Main Strengths:**

The idea is simple, and the theoretical analysis seems to be correct. Also numerical experiments are logical and confirm that new methods outperform the standard ones.

**Q4 Main Weakness:**

The final "Riemannian" formula is just gradient scaling by the norm of the weights squared. Seems to be quite simple, and it is not 100% clear why it should be better for generalization than the standard "learning rate schedules".  A very simple synthetic example could be useful. Also, can this scaling be applied even for non-PSI models?

The difference between the present work and the work of Cho is not very vivid: indeed, it covers some analysis and experiments, and also proposed a method, but conceptually it is a follow-up paper.

The SGDM has no theoretical justification.



**Q5 Detailed Comments To The Authors:**

Not much to say additionally compared to the comments in Q3 and Q4. The paper is conceptually very simple, easy to read and gives a simple recipe for the new SGD-type method by scaling gradients by the norm of the weights squared.
The question is how specific this scaling is to PSI. If this rule is heuristically used for other tasks? How it is related to batch norm actually in this sense? A small additional example that shows that the source is indeed in the manifold but not in a "magic scaling rule" would be great. Otherwise, a reasonable paper.

**Q7 Justification For Your Score:**

A reasonable paper: simple idea, good experiments. No strong objections.

**Q9 Complying With Reviewing Instructions:**

1: Yes.

---

### Official Review · Reviewer_aC9c · 2022-04-13

**Q2(1) Originality/Novelty:** 2
**Q2(2) Significance/Impact:** 2
**Q2(3) Correctness/Technical Quality:** 3
**Q2(6) Clarity Of Writing:** 2
**Q6 Overall Score:** 4
**Q8 Confidence In Your Score:** 4

**Q1 Summary And Contributions:**

This paper presents a method for efficiently training networks with batch norm (and presumably also layer norm, weight norm, etc.). It's based on optimizing over the quotient manifold, based on the positive scale invariant property of normalization layers. The main contribution is theoretical convergence results. Experiments also show good practical performance on neural net training.

**Q2 Assessment Of The Paper:**

More detailed information regarding each of these aspects is given below:

**Q2(4) Quality Of Experiments (Optional):**

3: Good: The experimental evaluation is adequate, and the results convincingly support the main claims.

**Q2(5) Reproducibility:**

3: Good: Key resources (e.g., proofs, code, data) are available and key details (e.g., proofs, experimental setup) are sufficiently well-described for competent researchers to confidently reproduce the main results.

**Q3 Main Strengths:**

The paper is well written: it clearly explains the background on manifolds, the proposed method, etc. The method is appealing and simple to implement, and it seems to work well in practice.

**Q4 Main Weakness:**

My main source of hesitation is that the relationship to past work is not clearly delineated. The paper cites Cho and Lee (2017) and Huang et al. (2017); while I wasn't previously familiar with these works, they seem very similar to the proposed approach. This submission claims the main limitation of those works is that they didn't prove convergence rates; does this mean this submission is primarily about proving convergence rates for existing methods?  The similarities and differences need to be made clear, and perhaps direct experimental comparisons are needed.

I'm also not clear on the novelty and significance of the convergence rate results which, from a glance, look generic. The proofs are in the appendix; a sketch in the main body would be helpful in getting across the key ideas.

If the proposed method is equivalent to an existing method, what is the added value of the experiments?

**Q5 Detailed Comments To The Authors:**

The implicit learning rate decay is described here as an "adverse effect" (p. 1), but this seems context-dependent. If memory serves me, the implicit decay effect serves as a sort of 1/sqrt(T) decay, which is good in contexts like online convex optimization. This submission also explicitly introduces a 1/sqrt(T) decay for PSI-SGD; isn't this sort of like manually reintroducing the BN implicit decay effect?

Minor:

p. 1, "These networks can converge to different local optima": you mean non-equivalent local optima (i.e. representing different functions), right?

I'm not sure Figure 1 adds much value.

p. 7: "Figure ??"

In the CIFAR experiments, how were the decay schedules chosen?  Since the main difference between PSI-SGD and ordinary SGD is the absence of implicit learning rate decay, it seems like the optimal explicit decay schedule could be very different between them (and hence an arbitrary choice could favor one or the other).


**Q7 Justification For Your Score:**

My main hesitation about this paper is about the relationship to prior work. If the rebuttal gives a good answer to this, I'll raise my score.


**Q9 Complying With Reviewing Instructions:**

1: Yes.

---

### Decision · Program_Chairs · 2022-05-15

**Decision:**

Accept (Poster)

**Comment:**

Meta Review: The paper studies the problem of how to improve the optimization stability of batch normalization. Algorithms are proposed to ensure that every group of equivalent weights will converge to the equivalent optima. Good performance has been observed for the proposed methods. Overall, the technical contributions of the paper are inspiring. We ask the authors to further improve the presentation of the paper in the final version.